# Study of The Impact of Users’ Features on Dimensional Allowances Resulting from the Use of Personal Protective Equipment

**DOI:** 10.3390/ijerph20043380

**Published:** 2023-02-15

**Authors:** Joanna Szkudlarek, Grzegorz Owczarek, Marcin Jachowicz, Bartłomiej Zagrodny, Jędrzej Sencerek

**Affiliations:** 1Department of Personal Protective Equipment, Central Institute for Labour Protection-National Research Institute, 48 Wierzbowa Street, 90-133 Lodz, Poland; 2Department of Automation, Biomechanics and Mechatronics, Lodz University of Technology, 1/15 Stefanowskiego Street, 90-924 Lodz, Poland

**Keywords:** personal protective equipment, dimensional allowances, ergonomic design, occupational health and safety

## Abstract

Up-to-date anthropometric data on the human population are needed for designing safe and ergonomically efficient workplaces. An important determinant of safety and ergonomic comfort at work is knowledge of the value of dimensional allowances (DAs) when using personal protective equipment (PPE) as the dimensions and space occupied by workers increase. This is particularly important in environments characterized by spatial constraints. However, it is not well known to what extent the aforementioned DAs are affected by the users’ features. The anthropometric dimensions of 200 people (151 males and 49 females) were obtained from 3D scans, and these became the basis for calculating DAs when using PPE kits normally worn by rescue and technical workers. DAs were determined for the entire body shape of a person wearing three types of PPE kits designed for firefighters, mine rescuers, and welders. In the study, maximum and mean values of height, width, and circumference DAs were obtained. In addition, percentage dimensional increments (DIs) were calculated. A three-dimensional analysis of the human body with and without PPE, involving a 3D scanning methodology, was applied to address the research question. Test results clearly indicate that the values of DAs do not depend on the anthropometric features of users, such as sex, age, and body height percentile—they remain constant for a given type of PPE. The presented data are useful for designing PPE products as well as work tools and infrastructure, including machinery, devices, workstations, means of transport, interiors, and building equipment. The results of the presented study indicate that dimensional allowances play a significant role in interactions between persons wearing PPE and their work environments. The obtained results (DAs and percentage DIs) are included in a new anthropometric atlas of human measures developed by the CIOP-PIB in 2023.

## 1. Introduction

There are two types of dimensional allowances (DAs), which should be distinguished as they are related to two separate areas:-DAs used for designing PPE characterized by a proper fit to the user’s body features; -DAs arising from the use of PPE that makes human–work environment interactions safer and more comfortable. 

DAs of the first type have been widely described as so-called clothing DAs. Such DAs are defined as the space between the human body and the garment, and they determine clothing fit and comfort of use, including heat transport, steam and moisture removal, and mobility. DAs of the second type are crucial for human interactions with the work environment in terms of allowing safe and unconstrained access of the human body or its parts to spatially restricted areas, such as manholes and ventilation openings [1,2,3]. Here, DAs are understood as differences between the dimensions of the human body and the maximum external dimensions of individuals wearing PPE (after adding the structural dimensions of the PPE). Already, in the 2001 book *Atlas of Human Measures* [4], it was noted that DAs affect occupational safety and health, and that improving the ergonomics of the PPE, tools, machinery, and workplaces has a bearing on work optimization.

There are two areas of human occupational activity where information about DAs is particularly relevant: technical and rescue operations in constrained and enclosed spaces, such as tanks, tunnels, ventilation ducts, etc. Among professionals working in these special conditions are welders, installers, service workers, electricians, mechanics, as well as firefighters, mine rescuers, and paramedics.

Work in enclosed and constrained spaces involves multiple hazards due to the additional risk of getting stuck. Most of these worksites fall into the category of difficult working conditions. This problem has been addressed by many authors [5,6,7,8]. Under these circumstances, workers may use a variety of PPE kits incorporating integrated respiratory protection systems, eye, face, and head protection devices, as well as protective clothing and gloves (depending on the particular hazards present). PPE kits as a whole occupy a certain space that must be accounted for in the form of DA values.

The structural features of PPE devices and workplace environments can be modeled using computer-aided design (CAD) software. The design process in a virtual environment involves the use of data from anthropometric databases and atlases. Anthropometric information has become an important source of input data for the production of PPE that is optimally “fitted” in terms of protective properties and user comfort. The inclusion of DAs arising from PPE use in databases will facilitate the design of PPE, as well as that of tools, machinery, and workplaces, with a view to optimizing their parameters, ergonomics, and occupational safety. Conversely, the maladjustment of workplace spatial parameters to the operator decreases the latter’s productivity and efficiency, and often leads to occupational illnesses.

In addition to drawing, CAD software can be used for simulating, modeling, and calculating the parameters of the object being designed [9]. CAD programs can be used to generate 3D human models, known as dummies, on the basis of data from anthropometric databases. Initially, such dummies were very similar to manikins, with the various segments of the dummy represented by simple geometric solids (e.g., in programs such as APOLIN, SAMMIE, ADAPS, WERNER, COMBIMAN, Catia, and Solid Edge). DA data should be prepared in a form that is accessible and universal, so that they are suitable for different applications and computer programs. Then, these data can be used to estimate the minimum dimensions for safe work.

The objective of the present work was to determine the distribution of DA values in a large sample consisting of males and females aged 18–65 years divided into 5th, 50th, and 95th percentile groups in terms of body height and corresponding protective clothing sizes. The results are presented in the form of numerical data that reflects absolute and percentage differences between the size of the human body in underwear and when wearing PPE. Analysis of the influence of the participants’ features on DA magnitude is relevant for the design of ergonomic workplaces, machinery, and tools. In spaces with limited access (including confined spaces), knowledge of DAs will make it possible to precisely define the space occupied by PPE users, ensuring their safe interaction with their work environments. In the case of rescue teams, for instance, data on the space occupied by a person wearing a complete PPE kit will allow for precise planning, optimum evacuation routes, shortened evacuation times, and increased effectiveness of rescue operations. Given the above, knowledge of dimensional allowances can significantly improve work safety in many occupations, especially those involving high-risk activities. 

## 2. Materials and Methods

The study sampled a group of 200 volunteers who were professionally active in various occupations (blue-collar and white-collar workers). The participants’ group contained both females and males aged 18–65 years, subdivided into the 5th, 50th, and 95th percentile groups by body height. The adopted age intervals were: 18–27, 28–37, 38–47, 48–57, and 58–65 years. The sizes of the age groups are given in Figure 1. As per point 23 of the Declaration of Helsinki of 1975, revised in 2013, , we took into account the confidentiality of the participants’ personal information. The volunteers were informed in detail about the aim, scope, and procedure of the experiment (especially about the harmlessness of the tests). 

The sample consisted of 31.6% females and 68.4% males, with the largest age groups being 38–47 and 48–57-year-olds, each accounting for 24% of the sample.

The participants were assigned to three body height groups defined by the 5th, 50th, and 95th percentiles, with males and females considered separately. The percentile groups were determined on the basis of an analysis of data from an atlas of human measures [10] and literature data [11,12]. In addition, the height groups based on percentile classification were assigned to the size data advertised by the manufacturers of protective clothing. The structure of the sample in terms of sex and size categories is given in Figure 2.

DAs arising from the use of PPE were determined for males and females in the 5th, 50th, and 95th height percentile groups, measuring from 152.9 to 199.0 cm, wearing full PPE kits of three types: those designed for firefighters, mine rescuers, and welders. The PPE kits consisted of head, eye, and face protection devices, as well as a respirator (for mine rescuers), protective clothing (jacket and pants, and a protective apron for welders), and hand and foot protection devices. The components of protection kits can be divided into those generating height DAs (head and foot protection devices), as well as those generating width and circumference DAs (protective clothing). The structure of PPE kits and their constituent components are shown as cross-sections in Figure 3.

In the study, a handheld 3D Artec Eva scanner (Artec Group, Luxembourg) [13] was used in conjunction with CloudCompare software for data processing [14]. Height, width, and circumference DAs were determined on the basis of scans of participants in underwear and wearing PPE. For that purpose, maximum dimensions were measured for each PPE kit. 

Before dimensioning, cleaning and surface reconstruction operations were conducted for each of the scanned objects using MeshLab software.

Preliminary work in CloudCompare consisted of point cloud generation (*sample points on a mesh*), initial scan superimposition (*translate*/*rotate*), and final scan superimposition (*finely register already aligned entities*). Length, width, and circumference were measured as distances between reference points in the analyzed clouds using the *cross-section* tool in the part of the model selected for measurement. Moreover, for the circumference calculation, a 1 mm high clipping box was set to extract a single contour option. CloudCompare can measure maximum, average, and minimum distances using distance computation tools (*compute cloud*/*cloud distance*).

In the first step, landmarks for the maximum dimensions of the human body were identified. Height was defined as the distance between the base (floor) and the vertex point (*v*) in the midsagittal plane (i.e., the topmost point of the head positioned in the Frankfurt Plane [15]). Shoulder width (maximum shoulder span) was defined as the distance between the acromion (*a*) on each scapula (the outward ends of the scapular spines). Another width measure relevant for the determination of DAs adopted in our analysis was the maximum width of the upper body at shoulder level. 

Examples of outlines of DAs arising from the use of full welding, firefighting, and mine rescue PPE kits are given in Figure 3 in three planes: (a) frontal, (b) sagittal, and (c) transverse. 

## 3. Results

In a preliminary study, we tested scanner accuracy, as well as the quality and repeatability of results. The methods of taking anthropometric measurements and the procedures for determining measurement accuracy are defined in the standards EN ISO 20685:2019-01 [16] and PN-EN ISO 7250-1:2017-12. According to the requirements of the former standard, data obtained from digital measurements involving 3D scanners may be used for generating anthropometric databases pursuant to PN-EN ISO 15534-1:2014-04 [17], as long as their mean variation is not greater than those given in Table 4.10 of the mentioned standard.

The results were subjected to comparative analysis. To that end, two measures (i.e., body height (A) and head circumference (B)) were taken by means of the traditional (manual) method and a 3D imaging (digital) method. Manual measurements were taken using standard tools: a tape measure and a stadiometer integrated with scales. The devices were checked using metrologically calibrated tools. Digital measurements were then taken using CloudCompare software tools. Length and width, being linear dimensions, were measured as distances between reference points in the analyzed clouds using the *cross-section* tool, represented by a clipping box with three principal axes (x, y, z). For the circumference calculation, a 1 mm high clipping box (*cross-section* tool) was set in the measurement area. Then, information about the circumference was extracted and displayed in the Properties table by means of the Export Envelope button.

The sample size was *N* = 14. The calculated means and standard deviations were used for comparing manual and digital measurements in terms of body height and head circumference, with the results in Figure 4. The values are given in millimeters, with a standard deviation of 0.33 and 0.46 for body height and 0.24 and 0.15 for circumference (for manual and digital measurements, respectively).

In accordance with the requirements of PN-EN ISO 20685-1:2019, the difference between manual and digital measurements was calculated and compared with the maximum permissible mean difference. The obtained mean difference was 2 mm (vs. a threshold of ≤4 mm), while the difference for mean head circumference was 7 mm (vs. a threshold of ≤9 mm). These results show that the obtained human measures (height and head circumference) met the requirements of the standard PN-EN ISO 20685-1:2019 in terms of mean differences between manual and digital measurements. Consequently, the measurements were deemed to comply with accuracy requirements, permitting them to be published in databases complying with PN-EN ISO 15534-1:2014-04. DAs were calculated as the difference between the dimensions of individuals in underwear and the same individuals wearing PPE kits. Height DAs are understood as the difference in the Y axis (vertical/long axis of the body). It should be noted that the overall height DAs arising from the use of PPE consist of DAs for foot protection (protective shoes) and DAs for head protection (helmets).

Width and circumference DAs are differences between the dimensions of individuals in underwear and the same individuals wearing PPE kits in the X axis (the axis perpendicular to the vertical) at the level of the maximum width/circumference of the human body. These DAs result from the use of protective clothing. Mean and maximum DAs were analyzed. 

Table 1 presents mean and maximum height, width, and circumference DAs for the three studied PPE types designed for firefighters, mine rescuers, and welders. The presented data were calculated from the results obtained for the 200 study participants described in the previous chapter.

In the next step of the study, we analyzed sex effects on DAs. Table 2 shows a comparison of mean height, width, and circumference DAs resulting from the use of PPE for males and females. On this basis, we calculated sex differences in DAs for firefighting, mine rescue, and welding PPE.

Table 3 presents an analysis of the effects of the participants’ body heights (based on assignment to the 5th, 50th, and 95th percentile groups) on mean height, width, and maximum circumference DAs arising from the use of PPE.

Figure 5 shows a summary of the results in terms of mean DAs for the three groups of participants defined by the 5th, 50th, and 95th percentiles for each PPE kit. 

An analysis of the design software environment revealed that DAs should be expressed in absolute numerical values that correspond to dimensional differences (in cm or mm), as well as in percentages (as percentage dimensional increments(DIs)). In designing PPE and ergonomic work environments, DAs expressed in terms of percentages may be most useful for designers as they are directly related to the dimensions of the human body without referring to PPE. Table 4 presents mean and maximum height, width, and circumference DAs expressed as percentage DIs.

## 4. Discussion

The results presented in Table 1 show that the magnitude of dimensional allowances (DAs) depends on the type of PPE kit donned by the user, as well as the construction of its components. A comparison of the values obtained for firefighting, mine rescue, and welding PPE (see Table 1) indicates that those values are characteristic of a given PPE type and range from a few to approx. 15 cm for height (max. 15.50 cm), up to 25 cm for circumference DAs (in both cases for firefighting PPE). Taking into consideration that designers use DA information for a variety of applications, DA analysis was carried out for mean and maximum values. Mean height DAs arising from the use of head and foot protection devices were 7.02–13.40 cm, while mean width and circumference DAs arising from the use of protective clothing were 4.40–9.32 cm and 9.51–17.10 cm, respectively.

A comparison of the mean width and circumference DAs given in Table 2 indicates that sex does not affect the magnitude of DAs. The maximum difference between males and females was 0.55 cm (4.1%) for height DAs, 0.70 cm (7.4%) for width DAs, and 0.35 cm (2.0%) for circumference DAs.

A comparison of the mean DAs given in Table 3 shows that the body height of participants (divided into 5th, 50th, and 95th percentile groups) did not have any effect on DA magnitude.

The greatest absolute differences in mean DAs between the percentile groups (differences between the maximum and minimum mean values) reached 1.01 cm for circumference DAs, 0.83 cm for width DAs, and 0.36 cm for height DAs. The observed differences in DAs between the percentile groups were more than twice as large for width and circumference DAs, but they are still acceptable due to the unstable dimensional nature of textile products. The mean values presented in Figure 5 attest to the absence of a correlation between DA magnitude and body height. DAs were found to be similar for the three percentile groups for each type of the PPE kit.

The percentage dimensional increments (DIs) given in Table 4 indicate that the use of the studied PPE kits leads to increments in width by up to approx. 23%, in circumference by up to approx. 20%, and in height by up to approx. 8%. The mean and maximum percentage DIs for firefighting PPE were 7.68% and 9.81% (height), 17.76% and 23.96% (width), and 13.22% and 19.53% (circumference).

## 5. Conclusions

The results of the study indicate that dimensional allowances (DAs) play a significant role in interactions between persons wearing PPE and the work environment.

This paper was devoted to full-body DAs and considered various PPE kits, which differed in their construction. Recent work [18] discussed DAs for parts of the human body (protective gloves). We present the results in this article as part of a larger project entitled “The Portrait of Polish People PL2030—An Atlas of Anthropometric, Biomechanical and Sensory Data”. Moreover, a database of dimensional allowances has been posted online by the Central Institute for Labour Protection–National Research Institute [19]. The database consists of input files for designing software, as well as algorithms for estimating DAs (so-called DA calculators), a useful tool for creating more ergonomic workspaces.

The presented research results indicate that the values of DAs do not depend on the users’ features, such as sex and body height based on the 5th, 50th, and 95th percentile groups associated with PPE sizes. Instead, DAs depend on the PPE type, shape, and dimensions. The maximum DAs reached 15.50 cm for height and up to 25 cm for the largest observed circumference. Percentage dimensional increments (DIs) indicate that the use of the studied PPE kits leads to increases in the dimensions of up to approx. 23% (DI for width). This means that the use of PPE cannot be neglected when designing PPE, tools, machines, and buildings, or while planning rescue operations and more. 

The results of the work should be taken into account by professionals designing both occupational and non-occupational infrastructure. The main idea of the study was to draw the attention of PPE designers to the fact that protrusions can increase the space needed for work, and may create additional difficulties as well as hazardous situations. In addition, engineers designing workspaces should be aware of the external dimensions of a person with and without PPE, as accounting for the total DAs resulting from PPE will make workspaces safer and more comfortable. The limitation of this work was that a vast array of PPE types are used across the world, and not all could be included; in the future, the database should be updated with new PPE designs as, other types of PPE and other occupations should be explored.

## Figures and Tables

**Figure 1 ijerph-20-03380-f001:**
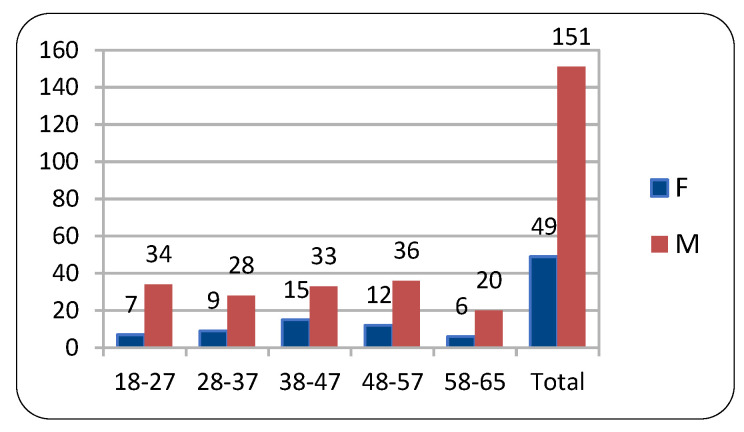
Sample distribution by age groups and sex.

**Figure 2 ijerph-20-03380-f002:**
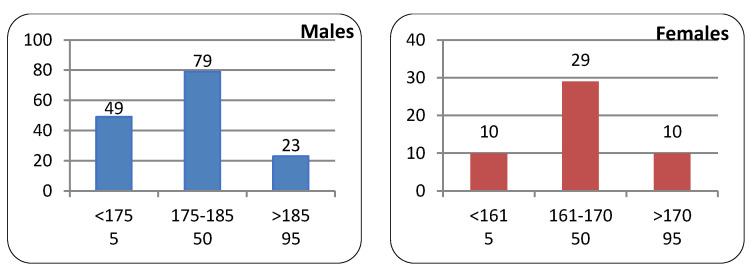
Sample structure in terms of height percentile groups and sex.

**Figure 3 ijerph-20-03380-f003:**
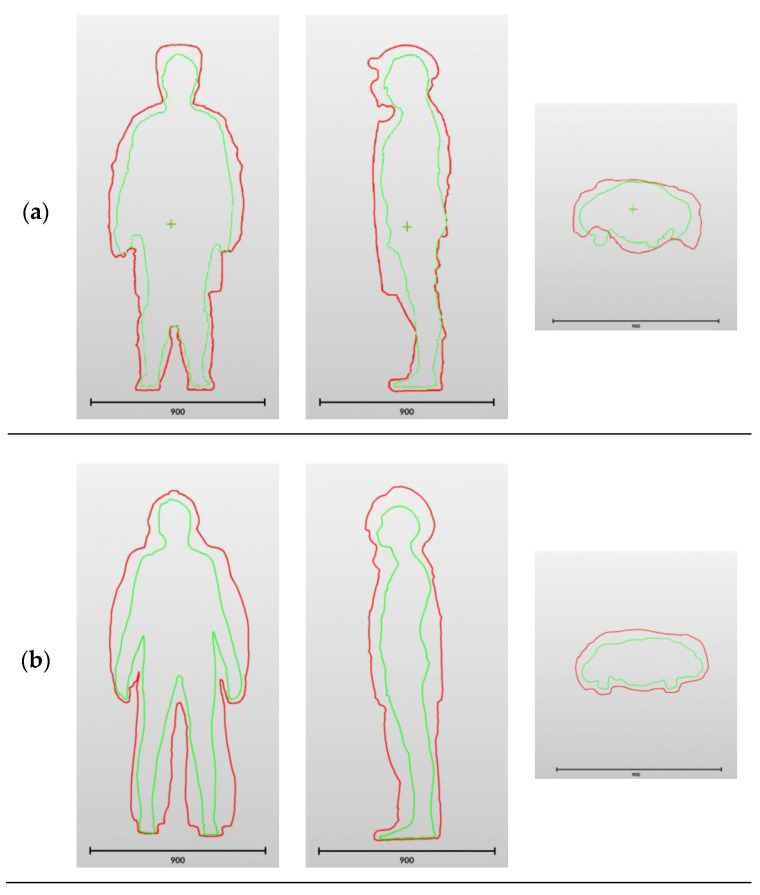
View of dimensional allowances arising from the use of full welding (**a**), firefighting (**b**), and mine rescue (**c**) PPE kits in the frontal plane (height and width DAs),the sagittal plane, and the transverse plane (circumference DAs at shoulder level).

**Figure 4 ijerph-20-03380-f004:**
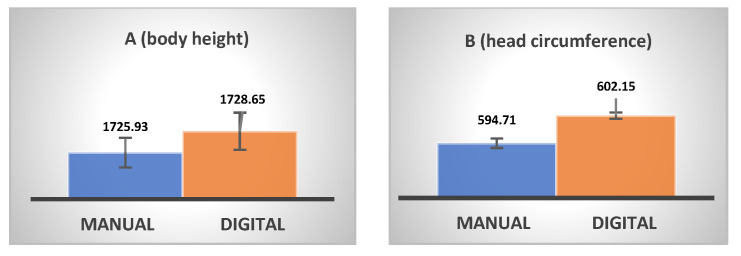
Comparison of manual and digital measurements in terms of body height (**A**) and head circumference (**B**).

**Figure 5 ijerph-20-03380-f005:**
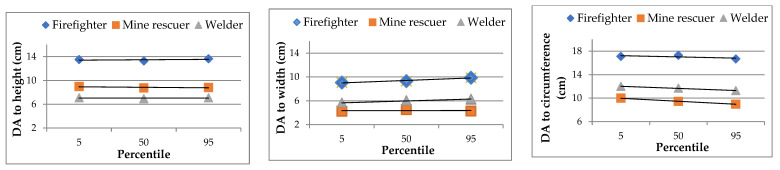
Mean DAs arising from the use of three types of PPE kits for three groups of participants defined on the basis of the 5th, 50th, and 95th percentiles.

**Table 1 ijerph-20-03380-t001:** Mean and maximum height, width, and circumference dimensional allowances (DAs) for firefighters, mine rescuers, and welders wearing personal protective equipment.

Types of Dimensional Allowances (DAs)	DAs (cm)					
	PPE Type					
	Firefighter		Mine Rescuer		Welder	
	Mean ± SD	Max.	Mean ± SD	Max.	Mean ± SD	Max.
Height DAs resulting from protective footwear	4.26 ± 0.14	5.50	3.51 ± 0.64	4.80	3.22 ± 0.42	4.60
Height DAs resulting from head protection products	9.15 ± 0.42	10.20	5.29 ± 0.71	6.50	3.80 ± 0.35	5.50
Total height DAs	13.40 ± 0.57	15.50	8.80 ± 0.07	11.00	7.02 ± 0.78	9.20
Width DAs	9.32 ± 0.71	13.50	4.40 ± 0,00	6.50	5.94 ± 1.06	8.50
Circumference DAs	17.10 ± 1.06	25.00	9.51 ± 1.35	14.00	11.70 ± 0.35	16.00

Notes: SD—standard deviation.

**Table 2 ijerph-20-03380-t002:** Mean height, width, and circumference dimensional allowances (DAs) for male (M) and female (F) firefighters, mine rescuers, and welders wearing personal protective equipment.

Types of Dimensional Allowances (DAs)	Mean DAs for Males and Females (cm)								
	PPE Type								
	Firefighter			Mine Rescuer			Welder		
	F	M	|F − M|	F	M	|F − M|	F	M	|F − M|
Height DAs	13.49 ± 0.21	13.33 ± 0.21	0.16	9.31 ± 0.49	8.76 ± 0.35	0.55	7.28 ± 0.07	6.94 ± 1.84	0.34
Width DAs	8.79 ± 2.47	9.49 ± 0,35	0.70	4.07 ± 1.06	4.51 ± 1.77	0.44	5.45 ± 1.77	6.10 ± 2.12	0.65
Circumference DAs	16.83 ± 2.83	17.18 ± 1.77	0.35	9.16 ± 2.83	9.62 ± 1.77	0.46	11.53 ± 3.54	11.75 ± 2.83	0.22

Notes: SD—standard deviation.

**Table 3 ijerph-20-03380-t003:** Mean height, width, and circumference dimensional allowances (DAs) with standard deviation (SD) for the 5th, 50th, and 95th percentile groups of participants wearing firefighting, mine rescue, and welding PPE kits.

DA Type	Percentile								
	5th			50th			95th		
	Firefighters	Mine Rescuers	Welders	Firefighters	Mine Rescuers	Welders	Firefighters	Mine Rescuers	Welders
Height DAs	13.50 ± 1.15	8.98 ± 0.21	7.08 ± 1.84	13.27 ± 0.28	8.73 ± 0.14	6.93 ± 0.95	13.63 ± 0.57	8.81 ± 0.14	7.07 ± 0.28
Width DAs	9.06 ± 2.12	4.25 ± 0.35	5.63 ± 0.00	9.30 ± 0.35	4.53 ± 1.06	6.02 ± 0.00	9.89 ± 1.77	4.28 ± 0.85	6.27 ± 2.12
Circumference DAs	17.09 ± 2.83	9.97 ± 2.12	11.99 ± 1.41	17.29 ± 1.41	9.44 ± 3.54	11.68 ± 1.77	16.67 ± 1.77	8.96 ± 2.83	11.27 ± 1.41

Notes: SD—standard deviation.

**Table 4 ijerph-20-03380-t004:** Maximum and mean height, width, and circumference DAs with standard deviation (SD) for females (in the 5th, 50th, and 95th percentile groups) in firefighting, mine rescue, and welding PPE kits expressed in terms of percentage dimensional increments (DIs).

Type of Dimensional Increment (DI)	Maximum and Mean DAs (%)					
	Firefighter		Mine Rescuer		Welder	
	Mean ± SD	Max.	Mean ± SD	Max.	Mean ± SD	Max.
Height DI	7.68 ± 0.42	9.81	5.04 ± 0.07	6.51	4.02 ± 0.78	5.63
Width DI	17.76 ± 0.71	23.96	8.39 ± 0.00	14.44	11.32 ± 1.06	16.67
Circumference DI	13.22 ± 1.06	19.53	7.32 ± 0.35	12.28	9.01 ± 0.35	13.60

Notes: SD—standard deviation.

## Data Availability

The data presented in this study are available on request from the corresponding author.

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
