# Peer review of "Study of The Impact of Users’ Features on Dimensional Allowances Resulting from the Use of Personal Protective Equipment"

_ijerph, 2023, doi:10.3390/ijerph20043380_

Round 1
Reviewer 1 Report
Summary: The study evaluated he impact of users’ features on dimensional allowances resulting from the use of personal protective equipment. The authors claimed that the value of DAs does not depend on the participants’ features, such as sex and body height. Instead, DAs depend on PPE type, shape, and dimensions.
Major comments:
1. They need to introduce DA and provide more background information about the topic to their readers. Without that piece of information, there will be no interest to finish reading their manuscript.
2. Authors did not even define their methods and how they calculated the DAs. They just reported the results without any details regarding their experiment (Also, no information regarding any controls in designing their experiment)
3. Finally, it would be great if authors can compare their results to previous similar studies in the literature and highlight the main difference/strengths of their study.
Author Response
Please find in the attachment.

Reviewer 2 Report
The authors of this paper present a study on the impact of PPE with regard to DAs determined based on the shape of the entire body. The work presented is sound and the conclusions are supported by the results.
My only concern is the fact that the authors do not provide the contribution of this specific paper with regard to other relevant contributions. In my opinion, the contribution of this paper should be included at the end of the Introductions in a concise manner.
Author Response
Please find in the attachment.

Reviewer 3 Report
The Abstract of the article is interesting, however it should be improved, It will be very important that the Abstract, in addition to the information they already contain, also include "What is the Research Question", which supports this research. That is, what was the reason that led the authors to develop this research. In addition, it will be important that the authors also include the Research Methodology that the authors followed in this research.
The authors in the introduction, write the following statement: “Work in enclosed and constrained spaces involves multiple hazards due to the additional risk of getting stuck. Most of such worksites fall in the category of difficult working conditions.”. It is a strong and relevant phrase for this research, so it would be very important for the authors to look for a literature reference, to support this statement!
The authors in the article, under "2. Materials and Methods", state that the sample was a group of 200 volunteers. However, it would be very relevant for the relevance of the research if, at this point, the authors indicated which were the criteria defined for the selection of the people who made up the sample.
The conclusions are very vague and should be reviewed and strengthened by the authors. It would be better if the authors put here some of the most important values obtained with this research, and reinforce in this way the importance and contribution of this research to the society and scientific community.
It would be very important that the authors at the end of the article indicate what future work can be developed from this research. It would be very good if this research did not stop here, given its potential, and that the scientific community could give it continuity, and give it the importance and relevance that this research may have in the future.
It would be very important that the authors at the end of the conclusions indicate the limitations of this research. All research has its limitations, and they are a natural part of research. However, the authors' sharing of their limitations helps the scientific community to understand the research and some of the results obtained.

Author Response
Please find in the attachment.

Reviewer 4 Report
In my opinion, the publication is interesting, although it requires corrections/supplements.
Detailed comments:
- Not reference to research achievements in this area, the introduction text is very laconic, the remaining part contains information important for conducting reliable research,
- What was the basis for selecting people to conduct the research, Was it random selection?, I think there is also some inconsistency here. The authors write about people employed in various positions (including white-collar) and then refer only to clearly indicated positions (firemen, mine rescuers and welders). There is also the question of what was the reason for choosing these professional groups,
- It would be worthwhile to indicate in the title the specificity of the problem, the research is limited to three groups only: firefighters, mine rescuers and welders.
Moreover:
- The colors in Figure 3 are illegible. In my opinion, the silhouette outline (green) should be replaced with another (e.g. blue) or the darkened area of the drawing should be abandoned,
- The literature is quite limited and does not indicate the scientific achievements in a given area, thus it would require supplementation and consideration in the introduction.
Author Response
Please find in the attachment.

Round 2
Reviewer 3 Report
The article has been corrected and improved by the authors, so congratulations!
Reviewer 4 Report
Thank you for considering the comments.
The changes of text make the paper is better and it can be published.